

# The effects of COVID-19 on agriculture supply chain, food security, and environment: a review

Haider Mahmood[1], Maham Furqan[2], Gowhar Meraj[3] and Muhammad Shahid Hassan[4]

[1] Department of Finance, College of Business Administration, Prince Sattam bin Abdulaziz University, Alkharj, Saudi Arabia
[2] College of Agricultural Sciences, Oregon State University, Corvallis, OR, United States of America
[3] Department of Ecosystem Studies, Graduate School of Agricultural and Life Sciences, The University of Tokyo, Bunkyo City, Tokyo, Japan
[4] Department of Economics and Statistics, Dr. Hassan Murad School of Management, University of Management and Technology, Lahore, Pakistan

Corresponding author
Haider Mahmood,
haidermahmood@hotmail.com

## ABSTRACT

COVID-19 has a deep impact on the economic, environmental, and social life of the global population. Particularly, it disturbed the entire agriculture supply chain due to a shortage of labor, travel restrictions, and changes in demand during lockdowns. Consequently, the world population faced food insecurity due to a reduction in food production and booming food prices. Low-income households face food security challenges because of limited income generation during the pandemic. Thus, there is a need to understand comprehensive strategies to meet the complex challenges faced by the food industry and marginalized people in developing countries. This research is intended to review the agricultural supply chain, global food security, and environmental dynamics of COVID-19 by exploring the most significant literature in this domain. Due to lockdowns and reduced industrial production, positive environmental effects are achieved through improved air and water quality and reduced noise pollution globally. However, negative environmental effects emerged due to increasing medical waste, packaging waste, and plastic pollution due to disruptions in recycling operations. There is extensive literature on the effects of COVID-19 on the environment and food security. This study is an effort to review the existing literature to understand the net effects of the pandemic on the environment and food security. The literature suggested adopting innovative policies and strategies to protect the global food supply chain and achieve economic recovery with environmental sustainability. For instance, food productivity should be increased by using modern agriculture technologies to ensure food security. The government should provide food to vulnerable populations during the pandemic. Trade restrictions should be removed for food trade to improve international collaboration for food security. On the environmental side, the government should increase recycling plants during the pandemic to control waste and plastic pollution.

# INTRODUCTION

The coronavirus 2019 (COVID-19) broke out in China (*Fulekar, 2020*) and rapidly spread all over the globe in a few months, which resulted in it being categorized as a pandemic by the World Health Organization (WHO) (*Arshad Ali et al., 2020*; *Green, 2020*). Incidentally, compared to less populous and wealthy countries, the dissemination of COVID-19 was found to be more common in extremely densely populated poor countries (*Dyer, 2020*), which affects the marginalized world population. The economic effect of this pandemic was observed in all sectors, including the agriculture sector, and caused food insecurity on a large scale (*Alabi & Ngwenyama, 2022*). For instance, COVID-19 decreased agriculture production due to lockdowns and other emergency measures (*Zhou, Li & Zhang, 2023*), and also reduced the financial capacity of the general population due to job losses and low income during the pandemic, which reduced the purchasing power of poor households to buy food and meet other basic needs (*Murakami, 2022*). Thus, COVID-19 resulted in increased global hunger and food insecurity (*Bloem & Farris, 2022*). The issue is not limited to the developing world alone, as it has become a major concern in high-income and developed countries as well.

There is a wide pool of literature, including studies, that talks about the negative effects of COVID-19 on food security across different parts of the world. In their global perspective on the matter, *Saboori et al. (2022)* talked about how the pandemic is more likely to harm food security in nations that are already facing such insecurities, and the negative effects come in the form of a lack of food availability as well as affordability. Additionally, the quality of food in these countries is also negatively affected by COVID-19. The pandemic did not affect natural resources. However, COVID-19 was seen to be responsible for the disturbance of all food supply chains, from production to retailing (*Picchioni, Po & Forsythe, 2021*). This disturbance in the food supply chain also led to price hikes in the food market (*Consoli et al., 2023*), which reduced the consumer's purchasing power.

Many other studies in the literature mentioned that COVID-19 is responsible for global food insecurity (*Song & Kim, 2022*), as it affects the availability, accessibility, usability, and sustainability of staple food items (*Guiné et al., 2021*). However, *Pakravan-Charvadeh et al. (2021)* claim that COVID-19 helped to raise food security in Iran due to government support and aid in various forms. *Mutegi et al. (2024)* argued in their research that the pandemic widely affected the food sector, and one of the effects was the less frequent use of high-quality fertilizers and seeds. In the challenging situation of the pandemic, lack of access to resources, and supply chain issues, smaller farmers were more likely to be negatively impacted than their larger counterparts. Their data from Kenya showed that average crop income for poor farmers declined a lot during the pandemic, while other indicators, including food quality, fertilizer use, overall calorie intake, and food consumption, also showed a decline. With multiple researchers arguing about the net effect of the pandemic on food security around the world, this topic needs further attention to understand various dimensions of the relationship between these two variables. The results of the analysis can help policymakers devise short-term and long-term policies to address food security issues across multiple income groups by facilitating the food supply chain industry.

After a discussion of the problems of the agriculture sector and food security issues during COVID-19, we discuss the environmental outcomes of this pandemic to provide a more holistic view of the matter. *Barman (2022)* stated that COVID-19 remains a blessing for the global environmental and ecological profile, as reduced transportation activities help reduce $CO_2$ emissions. New workplace policies and flexible work options like working-from-home (WFH) and an overall reduction in production due to less demand during COVID-19 led to less pollution from the transport sector. Similarly, water and solid pollution are also reduced, and global ecological systems remain less affected by commercial and production-related human activities (*Yuan et al., 2023*). The frequency of international and domestic flights also reduced dramatically, which reduced fuel usage and overall economic activities, resulting in lesser pollution (*Lo et al., 2022*). Thus, COVID-19 changed the social and economic lives of individuals due to different practices and policies in controlling the spread of the virus (*Lau, 2021*).

Reduced production, transportation, and other business activities help combat greenhouse gas (GHG) emissions significantly (*Mannarini et al., 2022*), at least during the peak phases of the coronavirus pandemic. On the demand side, COVID-19 disturbed the financial position of the world population due to lockdowns, less mobility, and other preventive measures, which resulted in lower demand for household, commercial, and industrial products. Eventually, demand-side factors affected the overall production and transportation of products (*Maijamaa, Nweze & Bagudu, 2020*). Hence, a one-quarter to one-third reduction in GHG emissions was observed in the midst of the pandemic in some of the highest $CO_2$-emitting countries (*Mannarini et al., 2022*).

Food security was negatively impacted by COVID-19 as a result of numerous supply-side and demand-side issues, and the literature demonstrates the connection between the two (*Picchioni, Goulao & Roberfroid, 2022*; *Louie, Shi & Allman-Farinelli, 2022*; *Pickering et al., 2023*; *Gebeyehu et al., 2023*). COVID-19 has a significant impact on the environment and agriculture supply chain, in addition to problems with food security. A thorough analysis of the body of research on COVID-19's impact on supply chains, food security, and the environment at large is lacking, despite the fact that investigations into these topics are still underway. Research on this topic is still evolving, and combining and synthesizing the literature can provide an opportunity to take a closer look at all the work being done.

In summary, the COVID-19 pandemic had extensive economic, environmental, and societal consequences. For instance, the agricultural sector has experienced substantial disruptions throughout its supply chain. Labor shortages, travel limitations, and unpredictable demand during lockdowns have hampered agricultural output and increased worldwide food insecurity, particularly among low-income households (*Mahmud & Riley, 2022*). Recognizing the importance of addressing these complex concerns, our research intends to review the studies on the pandemic's influence on agricultural supply networks, food security, and environmental sustainability. We hope to uncover viable measures to mitigate negative effects on the food industry by conducting an extensive literature review. We also look at the environmental implications, including both positive results like enhanced air and water quality and negative impacts like increased medical, healthcare, and

packaging waste. Our goal is to educate people about policies and measures that improve agricultural resilience, assure food security, and reduce environmental degradation.

The reviewed literature has been conducted before on the relationship between COVID-19 and food security (*Balgah et al., 2023*; *Bloem & Farris, 2022*; *Louie, Shi & Allman-Farinelli, 2022*; *Picchioni, Goulao & Roberfroid, 2022*; *Gebeyehu et al., 2023*). *La Rosa et al. (2020)* also conducted a review of the relationship between COVID-19 and the water environment. However, there is less work on comprehensively reviewing the relationship between COVID and the environment. So, we review the studies focusing on COVID-19's effects on air pollution, noise pollution, medical, chemical, and packaging wastes, and recycling. In addition, we also review the literature with a broader scope of food security, agriculture supply chain problems, and remedial measures during the pandemic. Thus, the present study conducted a comprehensive review to understand the dynamic relationships between COVID-19, food security, the agriculture supply chain, and the environment. For instance, we also review the suggested strategies provided in the literature to reduce food insecurity, as COVID-19 still exists around the globe. This analysis can help us understand the multifaceted effects of COVID-19 on creating a balance between food security and the environment. Therefore, this analysis provides useful insights for both scholars and policymakers, assisting in the establishment of food self-sufficiency programs through the optimal use of domestic resources or strategic food imports during times of crisis to ensure food security. Furthermore, policymakers might use the analysis presented in this study to solve issues in the food supply chain in the agriculture sector. Environmentalists will also benefit from the research findings, which will allow them to establish a balance between COVID-19's positive and negative environmental outcomes by targeting the reduction in negative environmental consequences of COVID-19.

## SURVEY METHODOLOGY

We explored Google Scholar in October 2023 to find the most relevant studies on the relationship between COVID-19, the agriculture supply chain, food security, and the environment for the period from 2019 to 2023. Moreover, some of the latest studies were also searched and included in the analysis in March 2024 to update the literature. We searched for ("COVID-19" OR "COVID") AND ("agriculture supply chain" OR "agriculture"). In the same way, we search for ("COVID-19" OR "COVID") AND ("food security" OR "food insecurity" OR "food"). Then, we select the most appropriate and relevant studies after reading the abstracts. We focus on the study with the scope of the effects of COVID-19 on the agriculture supply chain and food security, which include shortage of any kind of inputs during the pandemic, transport and trade restrictions, exports and imports restrictions during COVID-19, food production and waste problems, food poverty, food security remedies, or government support for food security during the pandemic. Then, we search for ("COVID-19" OR "COVID") AND ("environment" OR "pollution" OR "emissions" OR "air quality" OR "water quality" OR "waste"), and we focus on the study investigating the positive and/or negative environmental effects of COVID-19. We selected studies investigating all types of pollution, including air pollution,

water pollution, plastic pollution, medical and chemical waste pollution, and recycling concepts.

## LITERATURE REVIEW

This section comprehensively explores the literature on the effects of COVID-19 on the agriculture supply chain, food security, and environment. Moreover, this section is divided into four sub-sections to focus on each dimension of COVID-19. The first sub-section discusses the effect of COVID-19 on the agriculture supply chain. The second sub-section explores the effect of COVID-19 on food security, and the third sub-section discusses the remedial measures for food security. The fourth section displays the environmental pros and cons of COVID-19.

### The effect of COVID-19 on the agriculture supply chain

COVID-19 severely disturbed the agriculture market due to the shortage of labor and agricultural inputs (*Balgah et al., 2023*). The agriculture sector was disturbed as it hindered the supply chain from farmers to retailers (*Alabi & Ngwenyama, 2022*), resulting in worldwide food insecurity. The restrictions on travel and transportation also reduced the supply of farming inputs and reduced crop production. Moreover, the pandemic was also seen to be responsible for lowering overall household incomes, which reduced the demand for food (*Özkan, 2021*). Due to the lockdown, many food crop productions were reduced due to limited labor availability (*Stephens et al., 2022*), leading to food insecurity. On the other hand, due to transport restrictions in some parts of the world, large amounts of food were dumped and wasted (*Pisa, 2022*). Due to both lower food production and large food waste, food prices for staple food items, including dairy products, increased dramatically (*Cui et al., 2023*). Particularly, these price hikes were prominent in low-income countries due to disturbances in the whole food supply chain (*Soon-Sinclair, Nyarugwe & Jack, 2023*). Moreover, labor shortages due to sickness and the immobility of expatriate labor also reduced food processing units during COVID-19 (*Balgah et al., 2023*). The shape of the demand curve was changed due to fewer operational restaurants and changes in eating habits. Moreover, the disturbance of demand and supply in the food market generated buffer stocks, which increased food stocks (*Murti, 2021*).

*Balgah et al. (2023)* did a meta-analysis and stated that the pandemic had a severe impact on Africa's agricultural environment. Through a comprehensive assessment and synthesis of current data, their study shed light on the agriculture sector's multiple issues, which included disruptions across both supply chains and demand dynamics. For instance, industrial closures and transportation restrictions during the COVID-19 pandemic disrupted the global agriculture supply chain. Moreover, low income from lesser economic activities became a reason for low demand for products. Their comprehensive research presented persuasive evidence of the pandemic's far-reaching consequences, emphasizing the critical need for targeted actions to alleviate its negative effects on agricultural sustainability and food security across the continent. In another study, *Uyanga et al. (2024)* also conducted a similar analysis and indicated similar negative effects. In another study on the US and Mexico, *Quandt et al. (2022)* showed that in rural areas,

particularly, farmers were put under a lot of pressure during the pandemic as they had to jump through many hoops to get their work done in a timely as well as safe manner. The researchers interviewed 12 workers, and the results showed there could have been better resources available for these workers to deal with the stressors of the pandemic at work and home as they tried to take all the necessary precautions.

This section explains that the agriculture supply chain is disturbed due to lower production because of labor and agriculture input shortages during COVID-19. The shortage of inputs was observed due to travel restrictions and transportation limitations and resulted in a shortage of food, which led to a hike in food prices. The sickness of labor was also responsible for a reduction in food processing units. Moreover, perishable food was dumped due to a lack of storage facilities and restricted transportation during pandemic peaks. Thus, the supply shocks were responsible for food insecurity during the COVID-19 pandemic. Moreover, demand-side shocks were also observed in poor households due to limited income because of job losses and lower employment opportunities during lockdowns, which reduced their purchasing power to buy food. Moreover, some perishable food is also dumped and wasted due to transport restrictions in some parts of the world.

## Food insecurity during COVID-19

Food is a fundamental need of human health, and during COVID-19, it was particularly important to ensure food security and nutritional health for the masses (*Ben-Hamadou, 2021*). In the same manner, water is also a major necessity of life, and the literature argued that staying hydrated was crucial to ensuring a healthy immune system against the pandemic (*Fincher, Jepson & Connors, 2023*). Nevertheless, COVID-19 has disturbed food security to a great extent due to market disruptions, and it is difficult to ensure food security even with the production of basic food items due to labor shortages in the agriculture sector during peak COVID-19 periods. The pandemic disturbed food supply chains and food trade due to trade constraints (*Park & Liu, 2022*). For instance, the shutdown of international transport became responsible for reducing food imports and exports and disturbing the international food supply chain (*Jin et al., 2022*). *Yu-han et al. (2023)* did a simulation analysis and showed that declining remittance income also reduced food security. Moreover, food prices were also rising due to a shortage of food supply, which increases food insecurity. They suggested post-pandemic strategies to make the food industry more resilient so that sustainable growth could be ensured as the world recovers from COVID-19.

Food security depends on the amount and variety of food available in a country to meet the minimum dietary needs of society. COVID-19, along with other global problems, reduced food security. The transportation and travel restrictions due to COVID-19 severely impacted food security due to disturbances in the whole food supply chain, from production to consumers (*Belu, 2021*). Moreover, the improper government distribution system in times of food shortage and tight preventive COVID-19 policies made the issue of food insecurity more challenging (*Nagpaul, 2022*). Thus, food production declines along with trading problems led to food insecurity issues during the COVID-19 peaks. Side by side, low investment and production were observed due to limited government support during

the pandemic in low-income countries (*Gertz, 2021*). Due to low investment in the food sector, food in terms of agricultural products and meat was limited during the pandemic. Thus, the pandemic became a great threat to food security (*Martinez, 2021*). Moreover, food insecurity pushes more people to the poverty line in developing countries (*Yeboah, Shaik & Musah, 2021*). Millions of people were already suffering from poverty and food insecurity before COVID-19. However, COVID-19 created more food shortages, and food security became a problem for billions of people around the globe by putting more constraints on agricultural production (*Begg, 2020*).

Due to limited production and supply of food, food prices have jumped sharply, which has a great impact on the prices of non-food products as well (*Dyer, 2021*). In addition, developed countries restricted the food trade with developing countries for preventive measures (*Yu & Xiao, 2023*), which made the demand and supply of foodstuffs more volatile. Thus, importing restrictions would create food insecurity, which again gave rise to food prices, and households would have less food than they needed due to limited income during the pandemic (*O'Hara & Toussaint, 2021*). Particularly, poor households were affected as their income spending proportion on food was high compared to rich households. Moreover, low-income households are more associated with the agriculture sector in terms of their income. Thus, families involved in this sector were more affected by COVID-19 because of their income dependence on the agriculture sector (*Mahmud & Riley, 2022*). Even, agricultural products were not a source of COVID-19 spread (*Shahidi, 2020*).

Restaurants were another victim in the food sector, which is affected by social distancing. Thus, food demand by restaurants significantly declined during COVID-19 peaks (*Yacoub & ElHajjar, 2021*). The pandemic led to hikes in wheat and rice prices (*Haase, Zimmermann & Huss, 2023*), along with other necessary food items like cooking oil, vegetables, and fruits (*Lal et al., 2020*). The local production stoppage had a great impact on perishable food because it is expensive and difficult to store these goods in order to maintain a steady supply of perishable food. Thus, the stock of stored food did not match the needs of households during the pandemic (*Deaton & Deaton, 2020*). For instance, bulk-stored meat and some perishable vegetables expired during the peak of COVID-19. Thus, these food stocks were dumped into the soil, resulting in waste. Similarly, the milk was also dumped due to lower transportation during the pandemic (*Alabi & Ngwenyama, 2022*).

In a survey study, *Semakula et al. (2024)* investigated the households in Uganda during COVID-19 and found that 42.2% of households were worried about food security and 32.1% of households faced hunger. 25.2% of households skipped one meal, and 32.1% reduced their food consumption. *Sridhar et al. (2024)* explored a district of Zambia and found that most households reduced food consumption during the pandemic. However, the government or other agencies could help only 6.6% of households to support their dietary needs. *Nkoko, Cronje & Swanepoel (2024)* stated that the pandemic severely affected food insecurity among farming households in Lesotho, and household size, income, and education also affected food insecurity.

This section describes that some country-specific and regional studies confirmed food insecurity during the COVID-19 pandemic. Food insecurity stems from disturbances in

the whole food supply chain, from production to retailers, due to shortages of labor, disruptions in transportation, trade restrictions, improper government distribution systems, low investment in the food sector, and tight preventive COVID-19 policies. Particularly, restricted international transportation reduced the supply of imported food and upset the international food supply chain. Thus, food insecurity increased within nations reliant on imported food. On the other hand, some perishable food items were also wasted and dumped due to transport restrictions in some parts of the world. Thus, food insecurity increases hunger globally.

## Remedial measures for food security

Food security might be achieved with the concept of food self-sufficiency by using domestic resources (*de Paulo Farias & dos Santos Gomes, 2020*). Food self-sufficiency is meant to complete food production from local resources. However, food imports may also help in ensuring food security. The literature has shed light on the pros and cons of self-sufficiency (*Bikernieks, 2022*). During the international food crisis, *Clapp (2017)* explained the trade-off between self-sufficiency and international trade of food due to the volatile market during the crisis. The author argued that rejection of international trade would result in food insecurity. However, a balanced approach might be adopted to increase the capacity of domestic food production, along with some degree of freedom in food imports following international rules. In the same way, *Fontan Sers & Mughal (2020)* analyzed the concept of rice self-reliance in Africa during COVID-2019 and found that most rice has been produced domestically, but still, Africa depended on rice imports during the pandemic because COVID-19 reduced the consistency in the rice supply chain. Consequently, demand and supply could not reach equilibrium with locally produced rice. Moreover, rice price volatility during COVID-19 caused food insecurity among the African population. It was suggested to enhance innovation to increase agricultural productivity and to finance such innovation with government support.

In another study on the Gulf market, *Woertz (2020)* analyzed food self-sufficiency and security during COVID-19. Water scarcity was a major hurdle in the way of food self-sufficiency. Thus, the Gulf market depends on food imports to ensure food security. During the pandemic, it was also very important to ensure good management of food stocks. In this way, the Gulf market performed well in terms of food security. However, the marginalized groups of the population were affected due to their low income and high food prices. In the same way, *Deaton & Deaton (2021)* investigated Canada during COVID-19 and found that there was no significant problem with food security during COVID-19. Moreover, food prices were relatively stable compared to other parts of the world during COVID-19 in Canada. The shortage of food was covered with food imports and expat labor to enhance domestic food production. However, all kinds of foods were not in regular supply, which was managed by the households with a change in dietary habits. *Niles et al. (2024)* surveyed two states of the USA and found that home and wild food procurement helped to reduce food insecurity. On the other hand, some recent studies highlighted the importance of social protection programs, community-level interventions, and policy responses to reduce food insecurity during the pandemic (*de Haro Mota, Ortiz-Jiménez*

*& Blas-Yañez, 2024*; *Hangoma et al., 2024*; *Miller et al., 2024*). Moreover, *Orjiakor (2024)* stated that targeted supportive programs helped to reduce food insecurity during the pandemic.

The literature suggested some strategies for ensuring food security, including food self-reliance within a region and the need for a balanced approach between domestic production and international trade. For instance, increasing domestic resources for food production and reducing reliance on imported food may improve food security during any kind of pandemic. Moreover, wild food procurement also helped to reduce food insecurity. The food insecurity issue made us realize the importance of innovation, government support, and effective management of food stocks to address the multifaceted issues in the way of food security.

## The effects of COVID-19 on the environment

There can be a two-way association between COVID-19 and the environment. At first, the air is a source of virus and bacteria transfers among animals and human beings (*Zhou et al., 2020*). For instance, *Bossak & Andritsch (2022)* reported that high levels of Particulate Matter (PM) in the atmosphere would increase the spread of COVID-19. PM is a microorganism that is a composition of liquid and solid (*Prinz & Richter, 2020*), which is toxic to human health. PM can be inhaled easily and can be the cause of many respiratory diseases (*Zhang et al., 2020*). Side by side, PM, including PM2.5 and PM10, is a fine carrier of COVID-19, which can transfer COVID-19 disease from one person to another. Particularly, PM2.5 has more ability to enhance respiratory disorders and neurocognitive and cardiovascular illnesses (*Kandari & Kumar, 2021*).

The literature also reported a positive environmental effect of COVID-19. *Mandal & Pal (2020)* reported that PM10 has been reduced significantly in stone quarrying and crushing areas during COVID-19 due to lockdowns and lesser production activities. Similarly, *Lokhandwala & Gautam (2020)* investigated and found that PM2.5 declined significantly during COVID-19 lockdowns and other precautionary measures in India. *Mahato, Pal & Ghosh (2020)* also reported that both PM2.5 and PM10 declined due to lockdowns in India. *Pansini & Fornacca (2021)* stated that PM2.5 concentrations had significantly declined in some European countries and China. Similarly, PM2.5 concentrations declined in the USA. Other than PM concentrations, COVID-19 had a significant effect on reducing other dangerous gases, like NOx and carbon emissions, because of lockdowns and other COVID restrictions. Similarly, it reduced sulfur dioxide and carbon monoxide emissions (*Skiriene & Stasiškiene, 2021*). Even, $CO_2$ emissions and overall GHG emissions declined sharply during the first phase of COVID-19 due to the shutdown of industrial activities and transportation (*Mannarini et al., 2022*). Along with air pollution, COVID-19 also reduced global warming and humidity (*Lal et al., 2020*). *Marcucci et al. (2024)* investigated Norway and found that a free delivery strategy during the pandemic increased online shopping habits and reduced $CO_2$ emissions. *Damiani et al. (2024)* investigated Tokyo and found that COVID-19 reduced mobility and changed the lifestyle of the urban population, which helped to improve air quality. *Sahraei & Ziaei (2024)* investigated the EU and UK and stated that COVID-19 reduced $CO_2$ and $NO_2$ emissions. Compared to 2019, emissions were

reduced by 10.66% in 2020. *Lee, Kuwayama & FitzGibbon (2024)* investigated California and found that $NO_2$ emissions decreased by 54.2% on average in the year 2020 compared to 2019. This reduction of $NO_2$ was observed more in urban areas compared to rural areas. *Zeydan & Zeydan (2024)* investigated emissions from aviation in Turkey and found that a 33.8% to 50.3% reduction in different types of emissions in 2020 was observed compared to 2019, which was due to fewer local, international, and cargo flights.

China is the largest global $CO_2$ emitter due to heavy industrial and transportation activities. About half of GHG emissions in China declined during the first few COVID-19 phases because of fewer industrial and transportation activities (*Zhu et al., 2020*). Moreover, air quality is enhanced due to a reduction in PM concentrations and GHG emissions in England (*Villeneuve & Goldberg, 2021*). Furthermore, the ozone layer was improved during COVID-19 (*Baldwin & Lenton, 2020*; *Siciliano et al., 2020*). Furthermore, *Travaglio et al. (2021)* reported a significant reduction in GHG emissions in England. The shutdown of transportation alone led to a notable reduction in GHG emissions and PM concentrations due to a reduction in the consumption of petroleum products. Moreover, the industrial shutdown also contributed to a cleaner environment by reducing GHG emissions. *Singh & Mishra (2021)* found a half decline in pollution in Wuhan during COVID-19 compared to 2019. *Ma et al. (2023)* did analyses of a few months in China and found that $SO_2$ and CO declined significantly compared to the pre-COVID period in Shaanxi Province. Moreover, O3 concentration increased by more than 10% during the same period. Similarly, *Zhang et al. (2021)* found that O3 concentration increased significantly in the provinces of China due to precautionary measures for COVID-19 compared to pre-COVID periods. *Huang & Brown (2021)* found that pollution emissions declined in Germany. *Balakrishnan & Beemamol (2023)* analyzed the global $CO_2$ emissions and found a significant decline in these emissions compared to the pre-corona period, which might be due to the shutdown of national and international transportation.

Other than air quality, COVID-19 enhanced water quality due to limited industrial activities, urbanization, and the discharge of waste materials, crude oil, and plastic materials into water (*Rami, 2021*). *Roy, Ghosh & Roy (2021)* reported that the Ganga River got cleaner during the lockdown of COVID-19 in India. Suspended PM significantly reduced in water during COVID-19, which reduced water pollution. La Rosa et al. (2020) reported that China started filtering the water to stop the spread of COVID-19 by using chlorine. *Manoiu et al. (2022)* reported a significant decline in bio-chemicals in the water. Overall, water quality has improved during COVID-19 due to a reduction in water pollution, increasing rains, and the reduced use of water (*Wong et al., 2023*). In addition, the population of fish also increased due to fewer fishing activities and clean water (*Cooke et al., 2021*).

Along with other pollution controls, COVID-19 also had a significant positive effect on noise pollution. For instance, noise pollution from transportation and factories was reduced due to lockdowns due to precautionary measures for COVID-19. Thus, roadside noise pollution was reduced significantly in the cities during the COVID-19 period. Moreover, all economic activities were reduced, which resulted in all kinds of pollution, including noise pollution (*Espejo et al., 2020*). COVID-19 harmed forests as deforestation significantly increased to control COVID-19 spread (*de Oliveira, 2021*). However, COVID-19 also

had a positive effect on ecology by increasing biodiversity and reducing pollution due to precautionary measures (*Kalfatovic et al., 2020*). *Islam et al. (2024)* showed that significant improvements were seen in air quality during COVID-19. Reduced transportation led to a significant drop in pollution, but some of these effects did not last long. As urban activities started increasing, pollution levels started rising, while other issues, including medical waste and noise pollution, were on the rise.

Along with the positive environmental effects of COVID-19, some negative effects were also noticed in the literature. For instance, in the treatment of COVID-19 patients, medical waste has increased (*Tang, 2022*). Thus, medical waste is a significant challenge for waste management systems. *Maleki et al. (2024)* stated that medical waste increased in Iran during COVID-19. They suggested that a proper medical waste management system would help reduce pollution from medical waste. Moreover, the use of gloves and facemasks increased to control viral infection, which increased healthcare waste (*Fadare & Okoffo, 2020*). Moreover, due to home deliveries to discourage on-shop shopping to control the crowds in shopping places, the use of packing material increased, which increased waste (*Benson, Bassey & Palanisami, 2021*). This sudden jump in waste is a threat to the ecosystem, which needs urgent attention to increase recycling practices. However, recycling plants stopped operation during the pandemic times, which reduced the recycling of waste (*Argentiero, D'Amato & Zoli, 2022*) and posed a threat to the concept of the circular economy. Thus, it increased the plastic pollution from solid waste, healthcare waste, and medical waste (*Okoh, 2020*). Moreover, toxins from medical and chemical waste increased during COVID-19, which increased soil pollution (*Rezi & Rahayu, 2021*). Thus, the improper disposal of waste is a great threat to soil quality and biodiversity. So, there is an urgent need for collaborative efforts by the government and private sector to join hands against this soil and plastic pollution.

This section reviews the positive and negative environmental effects of COVID-19. The positive aspects include improvement in air quality by reducing PM concentration, $SO_2$, $NO_2$, NO, CO, $CO_2$ and GHG emissions due to the reduction in the use of petroleum products and due to declines in industrial, transport, and other urban activities during COVID-19 lockdowns. Moreover, water quality was improved due to a reduction in industrial activities, urbanization, wastewater, and the discharge of waste materials, crude oil, and plastic materials into water. The improved quality of the water also enhanced fish populations. In addition, noise pollution was reduced due to shrinking transportation and other urban activities. Thus, roadside noise pollution was controlled in cities. Further, COVID-19 also has a positive effect on ecology by increasing biodiversity. However, COVID-19 had some negative environmental consequences along with the mentioned positive effects. For instance, wastes from medical, healthcare, personal protective equipment, and packaging were increased, which resulted in soil and plastic pollution. Moreover, recycling plants were shut down due to the lockdown, which reduced the recycling of waste.

## CONCLUSION

This research reviewed the literature to understand the multifaceted effects of COVID-19 on the global food supply chain, food security, and the environment. Moreover, remedial measures to reduce food insecurity are also discussed. Thus, the present research contributes to the reviewed literature by investigating a comprehensive picture of the effects of COVID-19. The shortage of labor, travel restrictions, shortage of agriculture inputs, and changes in demand during lockdowns of the pandemic were responsible for disturbances in the global food supply chain, which was responsible for increasing food prices and food insecurity. Particularly, the disturbance of international transportation to protect against the spread of COVID-19 was responsible for lower exports and imports of food, which increased food insecurity in the regions with lesser food self-sufficiency. In addition, improper government distribution systems and tight preventive COVID-19 policies were also responsible for the food shortage. During COVID-19, low investments were observed in the food sector, which aggravated the threats to food security. Due to their perishable nature, foods expired during the peaks of the pandemic, which is dumped and responsible for food waste. The countries with lower levels of food self-sufficiency faced more problems of food insecurity because trade and travel restrictions created a severe shortage of food during peak periods of COVID-19. Thus, COVID-19 was responsible for increasing global hunger for billions of people. Disturbances in the global food supply were responsible for food insecurity during the pandemic. Moreover, the pandemic was also responsible for lower income due to job losses and fewer employment opportunities. Thus, lower income also reduced the demand for food and was responsible for food insecurity among poor households. Particularly, households associated with agriculture income were more affected due to disturbances in the agriculture market. In addition, the restaurant business was also disturbed by low food demand due to social distancing. The literature has suggested the adoption of innovative policies and strategies to protect the global food supply chain and a balanced approach that combines domestic production and food imports to achieve a sustainable global food system during the pandemic period.

On the environmental side, the pandemic improved air quality because of reductions in PM concentration, $SO_2$, $NO_2$, $NO$, $CO$, $CO_2$ and GHG emissions during lockdowns due to shutdowns or decreased activities in the industrial and transport sectors. The water quality was also improved due to a reduction in industrial activities and waste discharge during lockdowns, which also improved fish populations. Moreover, noise pollution was controlled due to less transport and other urban activities. Other than positive environmental effects, COVID-19 had negative environmental effects due to increasing medical, healthcare, personal protective equipment, and packaging waste. Thus, increased toxic elements from such waste enhanced soil pollution. Moreover, recycling plants were shut down during the pandemic, which intensified plastic pollution.

The reviewed literature provided some policy solutions to alleviate food insecurity and pollution during the pandemic. The government should invest in agriculture technologies to promote productivity and improve food production. For this purpose, the government should support R&D activities to develop agriculture technologies, genetic engineering,

and smart irrigation systems. These efforts would increase crop yield and agriculture productivity by reducing the use of agricultural inputs, which would increase sustainable food systems and also be equally beneficial in reducing environmental impact and climate change. Furthermore, these initiatives would increase employment in the rural sector, which could reduce rural poverty and promote rural development. On the demand side of food, the government should support the household income of vulnerable populations to enhance their purchasing power during a pandemic and should initiate some food assistance programs. The global food supply chain should be strengthened through international collaboration by removing trade restrictions for food items to ensure food security. Moreover, literature suggests the concept of food self-reliance to ensure food security within a region. Thus, local food production and international trade of food items should be balanced to avoid food insecurity in times of pandemic. To reduce waste generation during the pandemic, governments should adopt more recycling practices by enhancing recycling infrastructure, increasing public awareness campaigns, and incentivizing recycling programs. To reduce plastic pollution, the government should invest in biodegradable materials, reusable packaging options, and other eco-friendly packaging alternatives by promoting public–private partnerships. Such initiatives would reduce plastic pollution on the one hand and encourage innovative and sustainable packaging solutions on the other.

The present research could focus on the effects of COVID-19 on the agriculture supply chain, food security, and environment. However, future research may increase the scope of the research by investigating sustainable agricultural practices during the pandemic and by exploring the social and economic aspects of vulnerable agricultural communities.

### Funding

Prince Sattam bin Abdulaziz University funded this research work through the project number (2023/RV/03). The funders had no role in study design, data collection and analysis, decision to publish, or preparation of the manuscript.

### Grant Disclosures

The following grant information was disclosed by the authors:
Prince Sattam bin Abdulaziz University: 2023/RV/03.

### Competing Interests

Haider Mahmood is an Academic Editor for PeerJ.

### Author Contributions

- Haider Mahmood conceived and designed the experiments, performed the experiments, analyzed the data, authored or reviewed drafts of the article, and approved the final draft.
- Maham Furqan performed the experiments, authored or reviewed drafts of the article, and approved the final draft.

- Gowhar Meraj analyzed the data, authored or reviewed drafts of the article, and approved the final draft.
- Muhammad Shahid Hassan analyzed the data, authored or reviewed drafts of the article, and approved the final draft.

## Data Availability

This is a literature review.

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
