# Peer review of "The effects of COVID-19 on agriculture supply chain, food security, and environment: a review"

_PeerJ, doi:10.7717/peerj.17281_

## Round 0.1 · original submission · Major Revisions

Authors are advised to revise the manuscript as per suggestions of the reviewers.

Reviewer 1 ·

Basic reporting

The abstract section needs to add some policy discussions to handle the mentioned problems of COVID-19.
The introduction section of the paper should highlight the contribution of this review research to the existing body of review research.
The literature review section should start with the introduction of this section to elaborate on the subsequent sub-sections.
“supply chains and demand dynamics” mentioned in line 175 should further be explained.
“Labor shortages, travel limitations, and unpredictable demand during lockdowns have hampered agricultural output and increased worldwide food insecurity, particularly among low-income households” in 114-116 needs reference.
The reference Islam et. Al. (2024) in 353 should be reported like Islam et al. (2024).
The conclusion section should be expanded to discuss the all-important findings from the review articles. For instance, all the positive and negative consequences of COVID-19 on food security, the environment, and the agriculture sector should be discussed to display a comprehensive picture of the review.
The discussions of policies withdrawn from reviewed articles should be elaborated more.

Experimental design

Experimental design should be further explained.

Validity of the findings

The findings of the paper from the reviewed articles are well-explained. However, the abstract and the conclusion section need more attention to make them more informative and comprehensive.

Reviewer 2 ·

Basic reporting

The basic reporting is fine in this article. However, the introduction section should further explain the contribution of the study in the quoted studies in the last paragraph of the introduction section.

Experimental design

To ensure a comprehensive coverage of the diverse impacts of COVID-19 on agriculture supply chains, food security, and the environment, the survey methodology should further be explained in details to ensure a broad scope of literature inclusion to understand the topic.
I understand that the studies for review are collected for a certain period. But, still some latest studies on the topic published in 2024 may increase the scope of the study.

Validity of the findings

The negative environmental effects of COVID-19 discussed in the second last paragraph of the literature review section should further be explained.

Additional comments

This review article "The effects of COVID-19 on Agriculture Supply chain, food security, and Environment: a review" serves valuable insights by synthesizing existing literature on the multifaceted impacts of the pandemic on these critical domains. However, the following comments can improve the scientific rigor of the paper:
The findings of each sub-section of the literature review section can be further elaborated to convey the summary of the sub-section more comprehensively.
The policy discussions in the conclusion section should be further explained.
The contribution of this review research should be discussed in the conclusion section to highlight the importance of the study’s findings.
The future research directions should be added in the conclusion section, which should pave the way for future research on the topic.

·

Basic reporting

In the lines 26-27, the statement “Moreover, we investigate the environmental effects stemming from COVID-19.” is repeated. So remove it.
In the lines 33-35, the statement “The literature suggested adopting innovative policies and strategies to protect the global food supply chain and achieve economic recovery with environmental sustainability.”  is not enough. So, there is a need of adding some practical policies instead.
In the lines 60-61, the statement “Additionally, the quality and safety of food across these countries are also expected to be negatively affected even if natural resources are not affected by the pandemic.”  has no clear meaning and need rephrase to convey the context.
In the lines 84-86, the statement “Studies show that COVID-19 remains a blessing for the global environmental and ecological profile as reduced transportation activities help reduce CO2 emissions (Barman, 2022).” needs clarity as one citation is not enough to claims studies in literature.
In the lines 84-86, the statement “Reduced production, transportation, and other business activities help combat Greenhouse Gas (GHG) emissions significantly, at least for some time.” should be supported by a reference
I understand, the authors are summarizing some discussions held before. But still the arguments “the COVID-19 pandemic had extensive economic, environmental, and societal consequences. For instance, the agricultural sector has experienced substantial disruptions throughout its supply chain. Labor shortages, travel limitations, and unpredictable demand during lockdowns have hampered agricultural output and increased worldwide food insecurity, particularly among low-income households.” in lines 112-116 need reference to support.
In the lines 154-155-86, the statement“COVID-19 severely disturbed the labor and production market due to the shortage of labor and agricultural inputs.” need reference to support the argument.
In the lines 166-168, the statement “Moreover, labor shortages due to sickness and immobility of expatriate labor also reduced food processing units during COVID-19.” need reference to support the argument.
The statements in the lines 172-175 are not clear and need rephrasing.
In the lines 206-208, the statement “Yu-Han et al. (2023) did a simulation analysis and showed that declining remittance income also put pressure on food security which, in turn, put more pressure on food prices making it less than affordable for the public.” needs more clarity.
In the reference in line 242 “ElHajjar”, make sure the spelling of author right.
In the lines 316-317 “About half of GHG emissions declined during the first few COVID-19 phases because of fewer industrial and transportation activities (Zhu et al., 2020).”, it is not clear that reduction of emissions belong to china or all world.
In addition to above comments, many other sentences are hard to understand in all sections of the paper. So, manuscript requires a thorough language revisions to make research reader-friendly.

Experimental design

There is a need to put more details about survey methodology.

Validity of the findings

It looks fine.

Additional comments

The introductory statements must be added in starting literature review section.
The remdial policies in lines 251-278 of food insecuirty must have a separate sub-section in literature review section.
The novelty of the paper could be discussed in a better way in the last paragraph of the the introduction section.
The first paragrgh of the conclusion section needs more discussions to explain the all importamt dimensions of COVID’s effects.

---

## Round 0.2 · accepted · Accept

The paper is well-revised, therefore, I recommend that the manuscript be accepted for publication.

Reviewer 1 ·

Basic reporting

The authors incorporated all basic issues.

Experimental design

The study design of the revised version fulfills the required quality.

Validity of the findings

The validity of findings is now according to the required quality.

Additional comments

The revised version of this study is maintaining the quality of the journal.

Reviewer 2 ·

Basic reporting

The paper has been substantially revised and is ready for acceptance.

Experimental design

The methodology employed in this research is truly commendable, demonstrating a comprehensive and systematic approach towards addressing the research objectives and design.

Validity of the findings

The study findings are truly remarkable, unveiling valuable insights and offering significant contributions to the existing body of knowledge in the field.

Additional comments

Accepted.